# Silage Fermentation, Bacterial Community, and Aerobic Stability of Total Mixed Ration Containing Wet Corn Gluten Feed and Corn Stover Prepared with Different Additives

**DOI:** 10.3390/ani10101775

**Published:** 2020-10-01

**Authors:** Guangning Zhang, Xinpeng Fang, Guanzhi Feng, Yang Li, Yonggen Zhang

**Affiliations:** College of Animal Science and Technology, Northeast Agricultural University, Harbin 150030, China; zgn1234@126.com (G.Z.); dnfangxinpeng@163.com (X.F.); fengguanzhi123@163.com (G.F.)

**Keywords:** total mixed ration silage, additives, fermentation characteristics, microbial community, aerobic stability

## Abstract

**Simple Summary:**

Wet corn gluten feed (WCGF) is a feed containing high moisture and rapidly digestible, non-forage fiber and protein for dairy cows, that is difficult to preserve. The use of corn stover as roughage by ruminants is limited by its poor digestibility. Total mixed ration (TMR) silage is an ensiling mixed feed that can fully mix forage and concentrate in a specific ratio to satisfy the nutritional needs of dairy cows, which has become an effective method of preservation of high-moisture byproducts. The objective of this study was to investigate the effects of different additives on the fermentation quality, bacterial community, and aerobic stability of TMR silage containing WCGF and corn stover. Inoculation with lactic acid bacteria (LAB) + fibrolytic enzyme (EN) and LAB improved aerobic stability of TMR silages indicated by higher and more stable LA and AA contents, a smaller rise in pH, and yeast count than other silages. Total mixed ration silage inoculated the LAB + EN and LAB can become an effective method of preserving high-moisture WCGF and corn stover with poor digestibility.

**Abstract:**

The objective of this study was to investigate the effects of different additives on the fermentation quality, bacterial community, and aerobic stability of total mixed ration (TMR) silage containing wet corn gluten feed (WCGF) and corn stover. The TMR was ensiled with four treatments: (1) no additive (control); (2) lactic acid bacteria (LAB); (3) fibrolytic enzyme (EN); (4) LAB + EN. The EN and LAB + EN decreased the neutral detergent fiber and acid detergent fiber contents. Additives led to a higher lactic acid (LA) content (*p* < 0.0001) compared to control at all ensiling times. Silages inoculated with LAB and LAB + EN had higher dry matter (*p* = 0.0007), LA (*p* < 0.0001) and acetic acid (AA) contents (*p* < 0.0001) compared to control. The LAB and LAB + EN had significantly lowest ammonia nitrogen among the treatments, while no significant difference occurred after days 7 of ensiling. Silages treated with LAB and LAB + EN had a higher LAB count (*p* < 0.0001) and a lower pH, yeast, and mold counts compared to other silages. The LAB and LAB + EN greatly increased the portions of *Firmicutes* and *Lactobacillus* (*p* < 0.0001, and *p* < 0.0001, respectively) and reduced undesirable bacteria. Inoculation with LAB + EN and LAB improved aerobic stability of TMR silages indicated by higher and more stable LA and AA contents, smaller rise in pH, and yeast count than other silages. The LAB + EN and LAB reduced microbial diversity and improved the fermentation quality and aerobic stability of TMR silage containing WCGF and corn stover.

## 1. Introduction

The volume of imported high-quality roughage increases year by year, which causes higher feed costs in the dairy industry. Exploring new roughage sources is becoming a focus for both animal nutritionists and livestock producers. Wet corn gluten feed (WCGF), a rapidly digestible non-forage feed rich in fiber and protein coproduced in corn wet milling [1], has been applied as a potential feedstuff for ruminant animals. However, the high moisture content (36–40% DM) of WCGF can cause certain storage issues [2]. Traditionally, it is often dried for dry corn gluten feed, but the drying process wastes energy and produces air pollution problems. In addition, WCGF is often difficult to be ensiled alone because of inadequate fermentation, and it is due to its high moisture content and low sugar substrate levels. Combining WCGF with long fiber feeds with the small particle size can replace a portion of high-quality roughage in lactating cows. Enormous amounts of corn stover are produced annually worldwide, with a production of approximately 7 billion tons [3]. It is economical to convert corn stover into feedstuff, but the use of corn stover as roughage by ruminants is limited by its poor digestibility. Therefore, most corn stover is discarded, buried, or burned, resulting in resource waste and environmental problems due to unsuitable disposal methods.

A practice in Japan is to mix wet byproducts with dry feeds to prepare low-moisture total mixed ration (TMR) silage and this practice has gradually become an effective method of preserving high-moisture byproducts in animal feed utilized in many countries [4]. On the other hand, silage inoculated with *Lactobacillus plantarum* can make a dominant population in epiphytic microbes of ensiling material, which helps to produce sufficient lactic acid for rapidly decreasing of the pH value and helps to limit the activity of harmful bacteria [5]. However, silage inoculated with homofermentative lactic acid bacteria (LAB) had poor aerobic stability due to lower antimycotic volatile fatty acid [6]. A combination of *Lactobacillus buchneri* with *Lactobacillus plantarum* can produce high levels of acetic acid, improving the aerobic stability of silages [7]. The addition of corn stover will increase the fiber content and lower the digestibility of TMR silage. Combining inoculants with EN or EN alone has been reported to improve silage fermentation quality by promoting lignocellulose hydrolysis [4]. Additives, such as LAB, molasses are added to silage to dominate the natural population of bacteria in the forage and influence the dynamics of the microbial population during the ensiling process [8]. However, many bacterial species are viable but unculturable on selective media [9]. Culture-based techniques, characterization of denaturing gradient gel electrophoresis bands, PCR based Sanger sequencing techniques have been widely used to evaluate the bacterial population but underestimating the microbial community composition during ensiling [10,11,12]. Next-generation sequencing (NGS) techniques have revolutionized our understanding of bacterial communities in silage. Some studies have focused on the bacterial community of fresh and ensiled forages [13,14]. Thus, bacterial communities could reflect the silage properties and subsequent challenge during feeding. However, the microbial community related to TMR silage was rarely investigated by NGS.

The main objective of the present experiment was to examine the effects of LAB and EN on the fermentation quality, microbial community, and aerobic stability of TMR silage containing WCGF and corn stover.

## 2. Materials and Methods

### 2.1. TMR Silage Making

Corn stover corn silage and concentrate were obtained from a private dairy farm in Anda City, China (46.42° N, 125.33° E). The corn stover was chopped into approximately 2 cm pieces using a crop chopper. The concentrate was composed of ground corn, wheat bran, soybean meal, cottonseed meal, DDGS, and premix. Wet corn gluten feed was obtained from the Cargill Biochemical Co., Ltd., Songyuan, China. Ingredient percent of TMR was formulated to meet the animals’ requirements for nutrition based on the Cornell-Penn-Miner Dairy model version 3.08.01(University of Pennsylvania, Kennett Square; Cornell University, Ithaca, NY, USA; and William H. Miner Agricultural Research Institute, Chazy, NY, USA). The ingredient composition and contents of the TMR were as follows: corn stover (35.6% of DM), WCGF (8.59% of DM), corn silage (30.9% of DM), and concentrate (249% of DM). The chemical and microbial compositions of corn stover, WCGF, and TMR are listed in Table 1. The total mixed ration was ensiled with four different treatments: (1) no additive (control); (2) Lactic acid bacteria treatment (LAB; *L. plantarum* CGMCC10516 plus *L. buchneri* BNCC189797, applied at a ratio of 1:1; theoretical final application rate of 10^6^ colony-forming unite (cfu) /g of fresh matter (FM)); (3) 1 g fibrolytic enzyme per 1 kg FM (EN; 10,000 U/g activity, XS biotechnology Co., Ltd., Beijing, China); (4) LAB + EN. Additives were dissolved in 10 mL of distilled water and applied using a hand sprayer by spraying uniformly onto the mixture that was constantly hand mixed. The control was sprayed with an equal amount of distilled water. After thorough mixing, the TMR was ensiled in triplicate for each treatment at approximately 2 kg (fresh weight), followed by being tightly packed separately in a polythene bag (30 cm × 30 cm, Prodpad Biotechnology Co., Ltd, Chengdu, China) equipped with a hole that only enabled gas release, and sealed by using vacuum packing machine, there were 21 bags (4 ensiling days × 3 replicates + 3 days of aerobic exposure × 3 replicates) for each treatment. These TMR silages packed densities of 344 ± 4.41 kg DM/m^3^ were kept at ambient temperature (21–30 °C) for 3, 7, 15, and 30 days of ensiling. A total of 48 bags (4 treatments × 4 ensiling days × 3 replicates per treatment) were sampled for fermentation quality analysis. Total 36 bags (4 treatments × 3 days of aerobic exposure × 3 replicates per treatment) of 30 days were used for the analysis of aerobic stability.

### 2.2. Chemical Composition and Microbial Population Analysis

The pre-ensiled ingredients and TMR were divided into 2 subsamples and the ensiled TMR was divided into 4 subsamples. Dry matter (DM) content of the first subsample was determined by drying at 65 °C for 48 h in a forced-draft oven (DGX-9243B-1, Fuma Laboratory Co., Ltd., Shanghai, China). The dried sample was ground through a 1 mm screen in a microplant grinding machine (FZ102, Taisite Instrument Co., Ltd, Tianjin, China) and analyzed for DM and crude protein (CP) by methods 930.15 and 984.13 of the Association of Official Analytical Chemists [15]. Neutral detergent fiber (NDF) and acid detergent fiber (ADF) were analyzed according to methods described by Van Soest et al. [16]. Heat stable amylase and sodium sulfite were used for the NDF procedure and the results of NDF and ADF were expressed on a DM basis including residual ash. Water-soluble carbohydrates (WSC) were analyzed by the method described by Owens et al. [17]. The second subsample of 10 g was blended with 90 mL of sterilized water, and the extract was serially diluted in sterilized water. The population of lactic acid bacteria was measured by plate count on lactobacilli deMan, Rogosa and Sharpe (MRS) agar medium (Sinopharm Chemical Reagent Co., Ltd., Shanghai, China) incubated at 37 °C for 48 h under anaerobic conditions. Molds and yeasts were counted on potato dextrose agar medium (Sinopharm Chemical Reagent Co., Ltd., Shanghai, China) were kept in an incubator at 30 °C for 2–3 days. All microbial data were transformed to log10 and are presented on a wet weight basis. The third silage subsample (20 g) was diluted with 180 mL distilled water and stored in the refrigerator at 4 °C for 24 h, and then the extracts were filtered through two layers of cheesecloth. The filtrate was used for subsequent determination of pH, organic acids, and ammonia nitrogen (AN) contents. The pH was measured with a glass electrode pH meter (Sartorius Basic pH Meter, Goettingen, Germany). The lactic acid (LA), acetic acid (AA), propionic acid (PA), and butyric acid (BA) were measured by high-performance liquid chromatography (HPLC) [18]. Ammonia nitrogen was determined according to the indophenol-blue method [19].

### 2.3. Microbial Diversity Analysis

The remaining silage subsample (10 g) was mixed with 40 mL saline solution (NaCl, 0.90 g/g), and shaken at 120 r/m for 2 h. The filtered liquor through gauze was centrifuged at 10,000 r/m for 10 min at 4 °C. The supernatant was discarded, but the deposit was suspended in 3 mL saline solution. This procedure was repeated again for good precipitation. According to the manufacturer’s protocol, Genomic DNA was extracted using the TIANamp Bacteria DNA Kit (TIANGEN, Beijing, China). The extracted DNA was subjected to PCR amplification in triplicate using the Q5 High-Fidelity DNA Polymerase System (New England Biolabs, Ipswich, MA, USA). The V3–V4 region of the 16S rRNA gene was amplified using primers 338F (5′-ACTCCTRCGGGAGGCAGCAG-3′) and 806R (5′- GGACTACCVGGGTATCTAAT-3′). Purified DNA were sequenced using an Illumina MiSeq platform (Illumina, Inc., San Diego, CA, United States) at Baimaike Co., Ltd. (Beijing, China). The sequences obtained from the MiSeq platform were processed through open-source software pipeline QIIME (version 1.8.0, University of Colorado, Boulder, CO, USA) [20]. Alpha diversity indices (Chao1 and Shannon) were calculated by QIIME from rarefied samples using for richness and diversity indices of the bacterial community. Beta diversity was calculated using principal component analysis (PCA). Linear discriminant analysis (LDA) effect size (LefSe) analysis was performed online in the Galaxy online analysis platform (http://usegalaxy.org) to reveal the significant ranking of abundant modules in TMR silage groups. A size-effect threshold of 4.0 on the logarithmic LDA score was used for discriminative functional biomarkers. 

### 2.4. Aerobic Stability Analysis

After 30 days of ensiling, a total of 9 bags of each treatment were opened for aerobic stability test. The TMR silage of each bag was removed, thoroughly mixed, and loosely packed in 10 L plastic containers without compaction. Containers were stored at ambient temperature (20–25 °C) covered with a double layer of cheesecloth to avoid contamination and drying of the silage while allowing air to infiltrate the silage mass. Three bags of each treatment were sampled at 3, 6, and 9 days for fermentation index determination and microbial analyses after aerobic exposure.

### 2.5. Statistical Analyses

The chemical composition and microbial diversity data were subjected to one-way Analysis of variance (ANOVA) using the general linear model (GLM) procedure of SAS (version 9.3, SAS Institute Inc., Cary, NC, USA). The fermentation quality and aerobic stability were performed using the two-way ANOVA with the fixed effects of Treatment (LAB and EN), Day, and Treatment × Day, using the GLM procedure of SAS. Duncan’s test was employed for multiple comparisons, with the difference declared significant at *p* < 0.05.

## 3. Results

### 3.1. Characteristics of Fresh Materials and TMR before Ensiling 

The chemical composition and microbial counts of corn stover, WCGF, corn silage, and TMR are shown in Table 1. Corn stover and corn silage had a higher NDF and ADF contents and a lower CP content compared to WCGF. The moisture content of WCGF was higher than that of corn stover. Corn stover and WCGF had a lower WSC content. Epiphytic microbial community is another essential factor for silage fermentation, where >5.0 log cfu/g FM lactic acid bacteria at ensiling is necessary. However, Total mixed ration had a low (<5 lg cfu/g of FM) LAB count and a high yeast count (>5 lg cfu/g of FM). The WSC content (>5% of DM) of TMR was sufficient for fermentation during ensiling.

### 3.2. Chemical Compositions of the TMR Silages

As shown in Table 2, additions had no effect on CP content (*p* = 0.86), but the effect was significant (*p* < 0.05) on DM, NDF, ADF, and WSC contents. Silages treated with LAB and LAB + EN had higher DM content (*p* = 0.0007) compared to other treatments. Contents of NDF (*p* = 0.002) and ADF (*p* = 0.0003) in EN and LAB + EN silages were lower in the treated silages. Additionally, silages treated with LAB + EN had highest WSC content (*p* = 0.0005) in all silages.

### 3.3. Fermentation Quality of TMR Silages

As presented in Table 3, treatments, days of ensiling, and their interaction significantly (*p* < 0.05) affected the pH, LA, AA, and AN of TMR silages. Additives led to a lower pH value than that of the control at all ensiling times, and the combination of LAB and EN led to the greatest decline in pH among the treatments. The final pH values in LAB and LAB + EN treatment declined to below 4.2. Similarly, silages inoculated with EN also had significantly lower pH values (*p* < 0.0001) at all ensiling times compared to those of the control. The overall LA and AA concentrations of all TMR silages increased intensively over the entire ensiling process, but additives led to the highest LA concentration (*p* < 0.0001) compared to that of the control at all ensiling times. Silages inoculated with LAB had higher LA (*p* < 0.0001) and AA contents (*p* < 0.0001) compared to that of the control. Silages inoculated with EN had significantly higher LA (*p* < 0.0001) and lower AA (*p* < 0.0001) than the control. Silages inoculated with LAB + EN had the significantly (*p* < 0.0001) highest LA and AA concentrations in the entire ensiling process among the treatments. The overall AN concentration of all TMR silages increased intensively over the entire ensiling period. Additives decreased (*p* < 0.0001) AN level at all ensiling times compared to that of the control, and silages inoculated with LAB and LAB + EN had significantly lower AN among the treatments, while no significant difference occurred after days 7 of ensiling. Treatments, days of ensiling, and their interaction significantly (*p* < 0.0001) affected the populations of lactic acid bacteria, yeasts, and molds of TMR silages. All treatments increased the populations of LAB at early ensiling. Subsequently, the number decreased from 10^6^ to 10^8^ cfu/g FM^−1^. Silages treated with LAB and LAB + EN had a higher population of LAB (*p* < 0.0001) compared to other silages. LAB and LAB + EN silages had lower populations of yeast and mold than other silages (*p* < 0.0001).

### 3.4. Bacterial Community Analysis of Fresh TMR and TMR Silages

As shown in Table 4, the recovered readings ranged from 79,928 to 80,021 in all detected samples. The coverage values of all samples were approximately 0.99. A total of 1598 operational taxonomic units (OTUs) at the 3% dissimilarity level were determined to further analyze the bacterial community. The number of OTUs was lowest (*p* = 0.009) for the LAB + EN, EN, and control treatment in all treatments. TMR silages treated with LAB and LAB + EN had lower (*p* = 0.0002) Shannon and higher (*p* < 0.0001) Simpson values compared to other treatments. As shown in Figure 1a, the number of overlapping OTUs among the five treatments was 263 for the bacterial communities. The number of OTUs in the TMR silage was lower than that of the FM. As shown in Figure 1b, PCA clearly had the variance of the microbial community and revealed that the distance between the FM and the TMR silages was far. The FM and TMR silages can also be separated into different groups. The bacterial community compositions of the LAB + EN and LAB silages were most similar. The bacterial community compositions were in the same groups, while those of the other treatments were in different groups. The relative abundance of the bacterial community at the phylum and genus level is shown in Figure 2a,b. Furthermore, LefSe analysis performed to reveal the dominant biomarker genera of the bacteria community in all samples was shown in Figure 3a,b. The dominant phyla in the pre-ensiled TMR were *Firmicutes* (3.50%), *Bacteroidetes* (6.51%)*, Proteobacteria* (63.12%), and *Cyanobacteria* (21.13%), *Actinobacteria* (5.64%). The dominant genus of pre-ensiled TMR contained *Lactobacillus* (2.47%), *Sphingomonas* (3.22%), *Rahnella* (1.61%), *Pantoea* (6.59%), *Acetobacter* (16.7%), *Serratia* (6.26%), *Chryseobacterium* (1.42%), *Pseudomonas* (8.69%), *Rhizobium* (1.09%), and *Stenotrophomonas* (1.02%). However, the portions of *Firmicutes* and *Lactobacillus* increased greatly (*p* < 0.0001, and *p* < 0.0001, respectively) after ensiling, especially in the LAB and LAB + EN treatment silages (Figure 2c,g). Furthermore, the portions of *Cyanobacteria* and *Pseudomonas* decreased greatly (*p* = 0.02, and *p* = 0.0009, respectively) after ensiling (Figure 2e,h). *Bacteroidetes* was found to be the most abundant in the control silage (*p* = 0.04), with approximately 18.1% of the total population (Figure 2d). The silage treated with control, LAB, and LAB + EN, especially LAB, and LAB + EN decreased the relative abundance of *Proteobacteria* (*p* < 0.0001) compared with FM (Figure 2f). The silage treated with EN increased the relative abundance of *Rahnella* compared to other silages.

### 3.5. Aerobic Stability of TMR Silages

Fermentative characteristics and microbial compositions during aerobic exposure are presented in Table 5. Treatment, days of aerobic exposure, and their interaction had significant effects on the pH value, LA, AN, and AA concentration (*p* < 0.05). The pH value increased gradually in all treated silages during aerobic exposure, but the pH value of LAB and LAB + EN silages remained below 4.3 and that of LAB + EN silage had the lowest number among the silages during aerobic exposure. The pH of the control and EN silages gradually increased to more than 4.80 during aerobic exposure and rose by 0.54 and 0.64 after 9 days of aerobic exposure. The AA concentration in all silages steadily decreased in the aerobic exposure stage and that of the LAB and LAB + EN silages remained higher (*p* < 0.0001) than that of other silages during aerobic exposure. The LA concentration in LAB + EN silages had a slight increase during 6 days of aerobic exposure and that of the LAB silage gradually increased during 3 days of aerobic exposure and then remained constant until the end of aerobic exposure, whereas the LA concentration for the control and EN silages gradually decreased and always remained lowest during aerobic exposure. Ammonia nitrogen concentration increased gradually in all treated silages, along with an increase in pH during aerobic exposure. The concentration of AN gradually increased with aerobic exposure and AN concentration for LAB and LAB + EN silages remained lower (*p* < 0.0001) than that of other silages after 9 days of aerobic exposure. Treatment, days of aerobic exposure, and their interaction had significant effects on LAB, yeasts, and molds (*p* < 0.0001). There was a gradual decline in LAB numbers for all treatments and that of the LAB and LAB + EN silages remained higher (*p* < 0.0001) than that of the other silages during aerobic exposure. The levels of mold and yeast in all TMR silages gradually increased and that of the LAB and LAB + EN silages sharply increased to significantly (*p* < 0.0001) higher values than those of the control and EN silages after 9 days of aerobic exposure. In the present study, a lower amount of yeast was observed in LAB and LAB + EN silages during aerobic exposure, with a longer period of aerobic stability. Our results may due to high concentrations of AA decreasing the number of yeasts in the silages inoculated with LAB and LAB + EN.

## 4. Discussion

### 4.1. Characteristics of TMR before Ensiling

The WSC content of more than 50 g/kg of DM was crucial for assuring available fermentation quality [21]. The WSC contents in corn stover (2.12% of DM) and WCGF (4.26% of DM) were insufficient for adequate fermentation during ensiling without any additions during the ensiling process. However, the WSC content in TMR (55.3 g/kg of DM) was sufficient for fermentation during ensiling. Several studies reported that a population of the initial lactic acid bacteria of more than 10^5^ cfu/g FM^−1^ is considered a crucial factor in assuring the successful silage fermentation [22]. The content of yeast (10^5^–10^6^ cfu/g FM^−1^) was higher than that of LAB (10^2^–10^4^ cfu/g FM^−1^) in the TMR, which was insufficient for adequate fermentation.

### 4.2. Chemical Composition of TMR Silages 

Silages treated with LAB and LAB + EN had higher DM contents, which could be due to the inhibition of undesirable microorganism metabolism of silage nutrients by LAB and the combination of LAB with EN [4]. Yuan et al. [18] observed that inoculated LAB has higher fermentation efficiency than that of epiphytic LAB to transform sugars into lactic acid without producing secondary metabolites or gases. Fibrolytic enzyme is added to degrade silage cell walls to serve as a substrate to produce lactic acid leading to a decrease in pH to inhibit metabolism of undesirable microorganism [5]. Inoculation of LAB had some effect on fiber fraction by acid hydrolysis of hemicellulose [18]. Silages treated with LAB + EN had higher WSC content, which could be explained by the fact that more digestible plant cells were hydrolyzed by LAB producing the acid in the ensilage process [23] and EN degrading silage cell walls to increase the content of WSC [5].

### 4.3. Fermentative Quality of TMR Silages

A rapid decline in pH depends on LAB producing LA, which is important in reducing the growth of early undesirable microorganisms and the loss of nutrients [4]. Additives led to a lower pH and a higher lactic acid contents than that of the control at all ensiling times. The decline in pH values occurred during the entire ensiling process, which is in accordance with the report by Liu et al. [4] The final pH values in LAB and LAB + EN treatment declined to below 4.2. Similarly, silages inoculated with EN also had significantly lower pH and higher lactic acid concentration at all ensiling times compared to those of the control. The mechanism of action was different for the additives. Lactic acid, with strong acidity and a pKa of 3.86, is the main organic acid responsible for pH reduction [8]. With the addition of LAB, which drives a dominant population among the epiphytic microbes of the ensiling material, a wide variety of substrates can be fermented quickly, producing much LA [4]. The above results proved that LAB produced sufficient lactic acid and decreased pH during ensiling, which is information that has been widely applied in forage silage production [7]. Fibrolytic enzymes are added to degrade silage cell walls to serve as a substrate to produce lactic acid, and a combination of microbial inoculation and the addition of fibrolytic enzyme leads to a decrease in pH [5]. Similarly, Liu et al. [4] reported that the addition of the combination of inoculant and enzymes increases LA fermentation in TMR silage. TMR silage inoculated with LAB and LAB + EN had significant effects on AA concentration because of LAB contained *Lactobacillus buchneri* producing sufficient AA. *Lactobacillus buchneri* is a heterofermentative LAB, which is in accordance with the reports by Hu et al. [7] In addition, the highest AA concentration in the LAB and LAB + EN suggested that LA was converted into AA with prolonged ensilage because of addition of *Lactobacillus buchneri*, which is in accordance with the report by Alli et al. [24] Ammonia nitrogen in silage is also an important criterion for evaluating fermentation quality of silage, which is produced by clostridia bacteria and reflects the degradation levels of proteins and amino acids during ensiling [5]. Silages inoculated with LAB and LAB+ EN had significantly lower AN content among the treatments, while no significant difference occurred after days 7 of ensiling. The above results may be attributed to the aerobic microorganisms and plant enzymes rapidly inhibited by a sharp decline in pH, decreasing degradation of protein [5]. These results were in agreement with the study by Yuan et al. [18] who found that *Lactobacillus plantarum* and fibrolytic enzyme could suppress proteolysis and decrease AN content in TMR silage including rape straw. All treatments increased the populations of LAB at early ensiling. Subsequently, the number decreased 10^6^ to 10^8^ cfu/g FM^−1^. The above results were similar to the study showing that the number of LAB decreased as the ensiling duration of spent mushroom substrate increased [25]. Kim et al. [25] also found that the addition of LAB increased LAB populations on all ensiling days. Fibrolytic enzymes were added to degrade silage cell walls to serve as a substrate to promote a higher population of LAB [5]. The populations of yeast and mold were 3.0 log cfu/g and 2.0 log cfu/g at day 30 post-ensiling. Silages treated with LAB and LAB + EN had lower pH and antifungal high AA concentration inhibiting populations of yeast and mold [26].

### 4.4. Bacterial Community Analysis of Fresh TMR and TMR Silages

This study revealed the relative abundance and diversity of bacteria in fresh TMR and TMR silage treated with LAB and/or EN. NGS can provide a more detailed picture to exhibit the response of the bacterial community to silage status [8]. Analysis of the *α*-diversity of the bacterial community revealed the variance of the microbial community. The coverage values of all samples were approximately 0.99, indicating that most bacteria were detected. The number of OTUs in the TMR silage was lower than that of FM, indicating that the bacterial diversity of TMR decreased after ensiling. The number of OTUs was lowest for the LAB + EN treatment, indicating a small number of acidifying bacterial species in the TMR silage. Shannon and Simpson’s indices are used to measure species diversity. The larger the Shannon index, the smaller the Simpson index, indicated the higher species diversity of the sample. The TMR silages treated with LAB and LAB + EN had lower Shannon and higher Simpson values, indicating a small population of bacterial species in the silage. Analysis of the *β*-diversity of the bacterial community further elucidates the variance of the microbial community. The number of overlapping OTUs among the five treatments was 263 for the bacterial communities, reflecting that many similar microbes were involved in silage, even when different additives were used. The PCA clearly revealed the variance of the microbial community by the clear separation of the FM and the TMR silages, which indicated that the microbial communities of fresh TMR were significantly different from those of the TMR silages and that ensiling could be the main factor affecting fermentation. The dominant phyla in the pre-ensiled TMR were *Firmicutes, Proteobacteria, Bacteroidetes, Cyanobacteria,* and *Actinobacteria*, which were also detected in other material [23]. However, *Firmicutes* increased greatly and *Proteobacteria* decreased greatly after ensiling, especially in the LAB and LAB + EN silages. *Firmicutes* is well known to be the predominant phylum in the bacterial community involved in lactic acid fermentation in silage [12]. Approximately 74.1 ± 3.28% of the alfalfa silage bacterial community belonged to this phylum, and approximately 20.4 ± 1.65% belonged to the phylum *Proteobacteria* [23]. A similar study also found that the relative abundance of *Firmicutes* increased from 8.1 to 70.6%, whereas that of *Proteobacteria* reduced from 89.6 to 26.9% after 40 days of ensiling, indicating that fermentation involves a shift in the bacterial community from *Proteobacteria* to *Firmicutes* before and after ensiling alfalfa forage [27]. *Bacteroidetes* and *Cyanobacteria* are also important components of the bacterial communities in silage [23]. *Bacteroidetes* is mainly involved in the hydrolysis of complex macromolecular organic matter, such as the degradation of carbohydrates into monosaccharides [28]. Zhao et al. [29] observed that *Bacteroidetes* was negatively correlated with WSC content. WSC content of control silage was lower than that of other silages. Thus, the relative abundance of *Bacteroidetes* in control silage had the highest level. Li et al. [30] found that *Cyanobacteria* as the predominant microorganism before ensiling was mainly replaced by *Lactobacillus* and *Enterobacter* after ensiling. The dominant genus of pre-ensiled TMR contained *Lactobacillus, Sphingomonas, Rahnella, Pantoea, Acetobacter, Serratia, Chryseobacterium, Pseudomonas, Rhizobium, and Stenotrophomonas*, which was slightly different from other result, indicating the distribution of dominant genus was attributable to type of fermentative materials. The fact that the bacterial diversity and taxonomic composition of the fresh TMR differ from that of the TMR silage reflect the effects of the ensiling. The portion of *Lactobacillus* increased greatly after ensiling, especially in the LAB and LAB + EN silages. *Lactobacillus* is the main bacteria involved in lactic acid fermentation during ensiling [8]. Pang et al. [12] investigated the microbial population of naturally fermented silages and found that *Lactobacillus* was dominant on sorghum, forage paddy rice, and alfalfa silages. Lactic acid-producing cocci initiate lactic fermentation in the early stage of the ensiling process, while *Lactobacillus* plays an important role in pH reduction at the later stage [31]. Inoculation with LAB increased the relative abundance of *Lactobacillus*, which was probably due to the improvement of exogenous LAB than epiphytic LAB on fresh forage. The additive of EN degrades silage cell walls to increase the content of WSC increasing the relative abundance of the *Lactobacillus* [5]. The addition of LAB and LAB + EN usually initiates lactic acid fermentation at the early stages of ensiling, thereby stimulating the dominance of *Lactobacillus* species. The bacterial community compositions were in the same group, which indicated that the LAB + EN and LAB played a catalytic role in fermentation, promoting the growth of similar flora. Our results revealed that TMR silage decreased greatly the relative abundance of *Pseudomonas* after ensiling. However, Ogunade et al. [23] observed that the relative abundance of *Pseudomonas* increased in silage. The reduction in diversity may have been because additives reduced the pH by increasing lactate and acetate concentrations, respectively [8], which may have inhibited the growth of the other bacteria. Li et al. [32] observed that *Rahnella* was detected in untreated silage, whereas some of these bacteria disappeared or became faint with *L. rhamnosus* treatment. However, *Rahnella* was found to be the most abundant in EN silage, reaching approximately 18.85% of the total population. The reason for the above result needs further investigation.

### 4.5. Aerobic Stability of TMR Silages

Silages are oxidized by aerobic bacteria, yeasts, and mold, which increases ambient temperature and nutrient deterioration when exposed to air. Therefore, it is essential to monitor the dynamics of the fermentation product and microorganism to evaluate the aerobic stability of silage. The change of pH in TMR silage could be used as criteria for aerobic spoilage, and a pH value exceeding the initial pH value by 0.5 indicates aerobic deterioration during aerobic exposure [18]. A significant increase in pH was found in control and EN after 3 days of aerobic exposure. The pH value for control and EN silages rose by 0.48 and 0.61, respectively during the aerobic exposure. Yuan et al. [18] observed that silage with high LA and WSC was more aerobically unstable when the silages were exposed to air which may because of the supplying supply of potential sources of readily available substrate for the growth of undesirable bacterial. TMR silages treated with EN had a significant reduction of lactic acid contents during aerobic exposure. Similarly, Weinberg et al. [33] found that pea and wheat silages with cellulase and hemicellulose had higher concentrations of residual WSC and LA enhancing lactate and assimilating yeast and mold development in higher enzyme treatments upon exposure. To improve aerobic stability, Hu et al. [7] applied *Lactobacillus buchneri*, one type of heterofermentative LAB, that can result in high acetic acid concentration that have strong antifungal properties and improve the aerobic stability of silages. At the same time, *Lactobacillus buchneri* can convert LA to 1,2-propanediol as a coproduct of the conversion of LA to AA [34]. As expected, the LAB had a positive effect on improving aerobic stability, which was reflected in smaller changes of pH and lactic acid for the addition of LAB and LAB + EN during aerobic exposure. The lactic acid in TMR silage without LAB sharply decreased to a lower level compared to silage with LAB, which is likely explained by the antimicrobial effect of acetic acid. AN concentration in silage reflected the degree of protein degradation. Massive degradation of proteins severely impacted the utilization of nitrogen by ruminants. Ammonia nitrogen increased gradually in all treated silages, along with an increase in pH during aerobic exposure, which may be due to protein breakdown by undesirable bacteria during silages exposed to air. AN in LAB and LAB + EN silages remained low than that of other silages after 9 days of aerobic exposure, which may account for the higher levels of AA and low pH inhibiting undesirable microorganisms. Yeasts and mold at high levels (over 10^5^ cfu/g FW) are primarily responsible for aerobic spoilage during aerobic exposure, and aerobic stability of silage decreased exponentially with increasing levels of yeast regardless of additives [26]. Yeasts and molds can oxidize lactic acid and WSC, increasing ambient temperature and pH. when TMR silages exposed to air, the mold and yeast count in control and EN silages increased rapidly and exceeded 5.0 log_10_ cfu/g FW at 6 days of exposure to air, while that in LAB and LAB + EN remained constant and below 5.0 at 9 days of aerobic exposure. Antifungal activity of high AA concentration decreased in the amount of mold and yeast in the silages inoculated with LAB and LAB + EN, and this finding agrees with that of Yuan et al. [18]. In addition, low mold and yeast contents in TMR silages treated with LAB and LAB + EN were attributed to the smaller variation of pH reflecting relative aerobic stability.

## 5. Conclusions

Additives could improve the fermentation quality of TMR silage to different degrees. TMR silages treated with LAB and LAB + EN had better fermentation quality compared to other additives. TMR silages treated with LAB + EN and LAB could enhance the abundance of *Lactobacillus*, reduce the amounts of undesirable microorganisms, and improve aerobic stability. In summary, LAB + EN and LAB reduced microbial diversity and improved the fermentation quality and aerobic stability of TMR silage.

## Figures and Tables

**Figure 1 animals-10-01775-f001:**
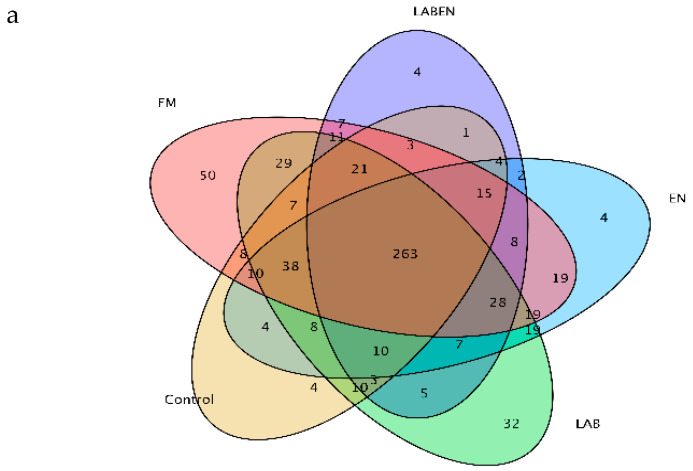
Venn analysis of operational taxonomic units (**a**) and the principal component analysis (**b**) of bacterial communities after 30 days of ensiling. FM, fresh material. Treatments, control, no additive; LAB, lactic acid bacteria; EN, 1 g fibrolytic enzyme per 1 kg FM; LABEN: combine lactic acid bacteria with 1 g fibrolytic enzyme per 1 kg FM.

**Figure 2 animals-10-01775-f002:**
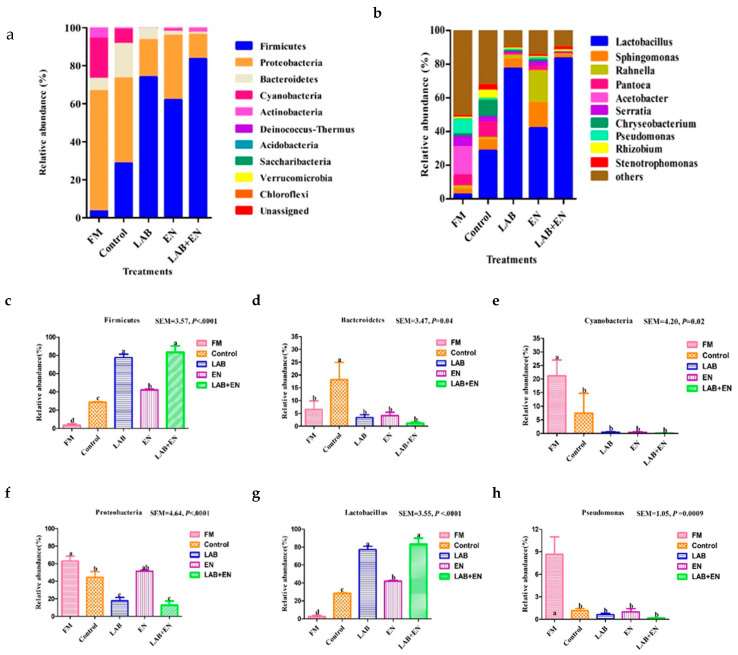
The microbial community of fresh materials and silages after 30 days of ensiling revealed by next-generation sequencing. (**a**), phylum; (**b**), genus level; (**c**) *Firmicutes*; (**d**) *Bacteroidetes*; (**e**) *Cyanobacteria*; (**f**) *Proteobacteria*; (**g**) *Lactobacillus*; (**h**) *Pseudomonas*. FM, fresh material; SEM, standard error of mean. Treatments, control, no additive; LAB, lactic acid bacteria; EN, 1 g fibrolytic enzyme per 1 kg FM; LAB + EN: combine lactic acid bacteria with 1 g fibrolytic enzyme per 1 kg FM.

**Figure 3 animals-10-01775-f003:**
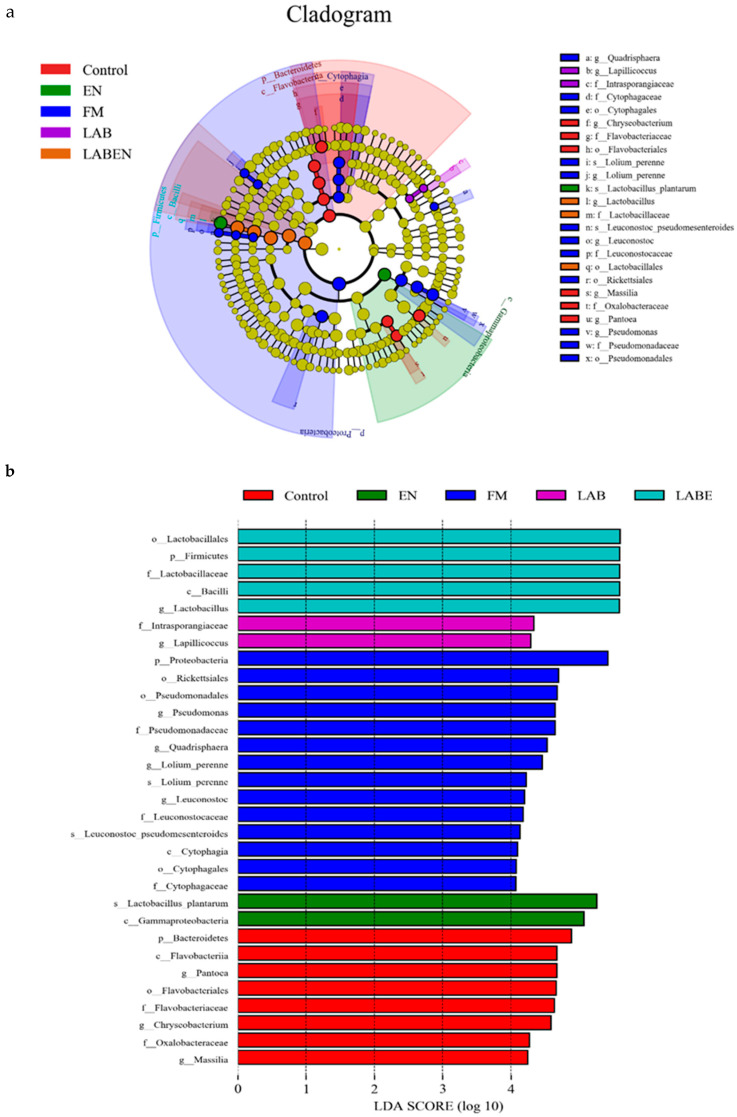
Comparison of microbial variations of fresh materials and silages after 30 days of ensiling using the Linear discriminant analysis effect size. (**a**) Cladogram; (**b**) Linear discriminant analysis scores. online tool. FM, fresh material. Treatments, control, no additive; LAB, lactic acid bacteria; EN, 1 g fibrolytic enzyme per 1 kg FM; LABEN or LABE: combine lactic acid bacteria with 1 g fibrolytic enzyme per 1 kg FM.

**Table 1 animals-10-01775-t001:** Chemical composition and microbial population of fresh materials and total mixed ration before ensiling. (mean ± standard deviation).

Items ^1^	Corn Stover	WCGF	Corn Silage	TMR
DM, % of FW	82.8 ± 0.32	61.4 ± 0.40	28.8 ± 0.31	44.6 ± 0.47
CP, % of DM	9.68 ± 0.17	22.2 ± 0.21	6.92 ± 0.06	12.2 ± 0.31
aNDF, % of DM	51.6 ± 0.67	36.5 ± 0.45	53.4 ± 1.90	48.5 ± 0.15
ADF, % of DM	46.6 ± 0.83	9.56 ± 0.12	30.5 ± 0.60	24.9 ± 0.32
WSC, % of DM	2.12 ± 0.09	4.26 ± 0.09	2.40 ± 0.26	5.53 ± 0.09
LAB (log cfu_10_/g FM)	3.64 ± 0.09	2.66 ± 0.05	7.96 ± 0.07	4.74 ± 0.06
Yeast (log cfu_10_/g FM)	6.74 ± 0.06	6.80 ± 0.08	2.97 ± 0.08	5.71 ± 0.07
Mold (log cfu_10_/g FM)	ND	ND	ND	2.68 ± 0.09

WCGF, wet corn gluten feed; TMR, total mixed ration. ^1^ DM, dry matter; FW, fresh weight; CP, crude protein; WSC, water-soluble carbohydrate; aNDF, neutral detergent fiber; ADF, acid detergent fiber; LAB, lactic acid bacteria; cfu, colony-forming units; ND, not detected.

**Table 2 animals-10-01775-t002:** Effects of lactic acid bacteria and/or fibrolytic enzyme on the chemical composition of total mixed ration silages containing wet corn gluten feed and corn stover.

Items ^1^	Control	LAB	EN	LAB + EN
DM, % of FW	42.1 ± 0.09 ^c^	43.4 ± 0.07 ^a^	42.6 ± 0.19 ^b^	43.5 ± 0.002 ^a^
CP, % of DM	13.2 ± 0.04	13.6 ± 0.005	14.1 ± 0.42	13.4 ± 0.06
aNDF, % of DM	43.5 ± 0.14 ^a^	42.5 ± 0.35 ^a^	40.6 ± 0.55 ^b^	39.6 ± 0.47 ^b^
ADF, % of DM	22.8 ± 0.10 ^a^	22.1 ± 0.08 ^b^	21.4 ± 0.27 ^c^	20.3 ± 0.05 ^d^
WSC, % of DM	2.06 ± 0.04 ^c^	2.59 ± 0.10 ^b^	2.78 ± 0.03 ^b^	3.82 ± 0.21 ^a^

^a–d^ Means in the same row followed by different letters differ significantly (*p* < 0.05). ^1^ DM, dry matter; FW, fresh weight; CP, crude protein; WSC, Water-soluble carbohydrate; aNDF, neutral detergent fiber; ADF, acid detergent fiber. FM, fresh material; control, no additive; LAB, lactic acid bacteria; EN, 1 g fibrolytic enzyme per 1 kg FM; LAB + EN, combine lactic acid bacteria with 1 g fibrolytic enzyme per 1 kg FM.

**Table 3 animals-10-01775-t003:** Effects of lactic acid bacteria and/or fibrolytic enzyme on the fermentation characteristics and microbial population of total mixed ration silages containing wet corn gluten feed and corn stover.

Items ^1^	Treatment ^2^	Days of Ensiling				SEM ^3^	*p*-Value ^4^		
		3	7	15	30		T	D	T × D
pH	control	5.20 ^Aa^	4.87 ^Ba^	4.74 ^Ca^	4.29 ^Da^	0.02	<0.0001	<0.0001	0.0001
	LAB	4.89 ^Ab^	4.76 ^Bb^	4.39 ^Cc^	4.18 ^Dc^				
	EN	4.93 ^Ab^	4.79 ^Bab^	4.61^Cb^	4.23 ^Db^				
	LAB + EN	4.85 ^Ab^	4.66 ^Bc^	4.35 ^Cc^	4.13 ^Dd^				
LA, % of DM	Control	3.64 ^Dc^	3.86 ^Cc^	4.78 ^Bc^	6.83 ^Ac^	0.10	<0.0001	<0.0001	<0.0001
	LAB	4.11 ^Dab^	4.94 ^Ca^	6.97 ^Ba^	8.53 ^Aa^				
	EN	3.87 ^Dbc^	4.55 ^Cb^	5.48 ^Bb^	7.99 ^Ab^				
	LAB + EN	4.29 ^Da^	5.06 ^Ca^	7.21 ^Ba^	8.74 ^Aa^				
AA, % of DM	Control	0.67 ^Db^	0.92 ^Cb^	1.37 ^Bc^	1.65 ^Ab^	0.04	<0.0001	<0.0001	<0.0001
	LAB	0.80 ^Da^	1.02 ^Ca^	2.11 ^Bb^	2.73 ^Aa^				
	EN	0.43 ^Dc^	0.86 ^Cb^	1.17 ^Bd^	1.33 ^Ac^				
	LAB + EN	0.85 ^Da^	1.11 ^Ca^	2.35 ^Ba^	2.89 ^Aa^				
AN, % of TN	Control	1.98 ^Da^	2.27 ^Ca^	2.60 ^Ba^	2.77 ^Aa^	0.04	<0.0001	<0.0001	0.0003
	LAB	1.58 ^Bbc^	1.79 ^Ab^	1.89 ^Ac^	1.92 ^Ac^				
	EN	1.70 ^Cb^	2.05 ^Ba^	2.18 ^Bb^	2.46 ^Ab^				
	LAB + EN	14.3 ^Bc^	16.5 ^Ab^	17.1 ^Ad^	17.1 ^Ad^				
LAB (log cfu_10_/g FM)	Control	8.70 ^Ac^	8.59 ^Bc^	8.30 ^Cc^	7.88 ^Dc^	0.20	<0.0001	<0.0001	<0.0001
	LAB	8.95 ^Aa^	8.90 ^Aa^	8.72 ^Ba^	8.41 ^Ca^				
	EN	8.81 ^Ab^	8.75 ^Ab^	8.54 ^Bb^	8.25 ^Cb^				
	LAB + EN	8.93 ^Aa^	8.85 ^Ba^	8.71 ^Ca^	8.35 ^Dab^				
Yeast (logcfu_10_/g FM)	Control	4.25 ^Aa^	3.27 ^Ba^	3.20 ^BCa^	3.15 ^Ca^	0.02	<0.0001	<0.0001	<0.0001
	LAB	3.98 ^Ac^	3.00 ^Bc^	2.89 ^Cc^	2.70 ^Dc^				
	EN	4.18 ^Ab^	3.16 ^Bb^	3.15 ^Bb^	3.08 ^Cb^				
	LAB + EN	4.02 ^Ac^	3.03 ^Bc^	2.86 ^Cc^	2.71 ^Dc^				
Mold (logcfu_10_/g FM)	Control	2.53 ^Aa^	2.40 ^Ba^	2.30 ^Ca^	2.15 ^Da^	0.02	<0.0001	<0.0001	0.66
	LAB	2.26 ^Ac^	2.15 ^Bc^	2.07 ^Cc^	1.92 ^Dc^				
	EN	2.42 ^Ab^	2.34 ^Bb^	2.23 ^Cb^	2.08 ^Db^				
	LAB + EN	2.23 ^Ac^	2.14 ^Bc^	2.08 ^Cc^	2.19 ^Dc^				

^a–d^ Means in the same column followed by different letters differ significantly (*p* < 0.05). ^A–D^ Means in the same row followed by different letters differ significantly (*p* < 0.05). ^1^ DM, dry matter; LA, lactic acid; AA, acetic acid; AN, ammonia nitrogen; TN, total nitrogen; LAB, lactic acid bacteria; FM, fresh material. ^2^ Treatments, control, no additive; LAB, lactic acid bacteria; EN, 1 g fibrolytic enzyme per 1 kg FM; LAB + EN, combine lactic acid bacteria with 1 g fibrolytic enzyme per 1 kg FM. ^3^ SEM, standard error of mean. ^4^ T, treatment; D, Days of ensiling; T × D, the interaction between treatment and days of ensiling.

**Table 4 animals-10-01775-t004:** Alpha diversity of bacterial diversity for fresh total mixed ration and total mixed ration silages of 30 days containing wet corn gluten feed and corn stover.

Items ^2^	Treatment ^1^					SEM ^3^	*p*-Value
	FM	control	LAB	EN	LAB + EN		
Reads	79,928	79,935	80,021	80,002	80,004	2449.4	0.46
OTU	385 ^a^	286 ^c^	353 ^ab^	313 ^bc^	261 ^c^	20.2	0.009
Chao1	432	361	409	345	340	30.2	0.20
Ace	425	353	405	345	355	26.7	0.20
Coverage	0.99	0.99	0.99	0.99	0.99	0.0001	0.45
Shannon	3.13 ^a^	3.17 ^a^	1.57 ^b^	2.58 ^a^	1.08 ^b^	0.233	0.0002
Simpson	0.12 ^b^	0.1 ^b^	0.57 ^a^	0.2 ^b^	0.67 ^a^	0.06	<0.0001

^a–c^ Means in the same row followed by different letters differ significantly (*p* < 0.05). ^1^ Treatments, control, no additive; LAB, lactic acid bacteria; EN, 1 g fibrolytic enzyme per 1 kg FM; LAB + EN, combine lactic acid bacteria with 1 g fibrolytic enzyme per 1 kg FM; FM, fresh material. ^2^ OTU, operational taxonomic units. ^3^ SEM, standard error of mean.

**Table 5 animals-10-01775-t005:** Effects of lactic acid bacteria and/or fibrolytic enzyme on the fermentative characteristics of the total mixed ration silages containing wet corn gluten feed and corn stover during aerobic exposure.

Items ^1^	Treatment ^2^	Days of Aerobic Exposure				SEM ^3^	*p*-Value ^4^		
		0	3	6	9		T	D	T × D
pH	Control	4.29 ^Da^	4.33 ^Cb^	4.59 ^Bb^	4.77 ^Ab^	0.009	<0.0001	<0.0001	<0.0001
	LAB	4.18 ^Cc^	4.23 ^Bc^	4.24 ^Bc^	4.28 ^Ac^				
	EN	4.23 ^Db^	4.41 ^Ca^	4.69 ^Ba^	4.84 ^Aa^				
	LAB + EN	4.13 ^Bd^	4.12 ^Bd^	4.20 ^Ad^	4.21 ^Ad^				
LA, % of DM	Control	6.83 ^Ac^	6.10 ^Bb^	5.55 ^Cc^	5.05 ^Dc^	0.13	<0.0001	<0.0001	<0.0001
	LAB	8.53 ^Ca^	9.95 ^Aa^	9.60 ^Ab^	9.10 ^Bb^				
	EN	8.00 ^Ab^	5.75 ^Bb^	5.20 ^Cc^	4.45 ^Dd^				
	LAB + EN	8.74 ^Ca^	10.4 ^Ba^	11.4 ^Aa^	10.3 ^Ba^				
AA, % of DM	Control	1.65 ^Ac^	1.55 ^Bc^	1.34 ^Cc^	1.18 ^Dc^	0.04	<0.0001	<0.0001	0.03
	LAB	3.39 ^ABb^	3.45 ^Ab^	3.23 ^BCb^	3.14 ^Cb^				
	EN	1.33 ^Ad^	1.17 ^Bd^	1.05 ^BCd^	0.99 ^Cd^				
	LAB + EN	3.83 ^Aa^	3.75 ^Aa^	3.52 ^Ba^	3.33 ^Ca^				
AN, % of TN	Control	2.76 ^Da^	2.88 ^Ca^	3.31 ^Bb^	3.76 ^Ab^	0.05	<0.0001	<0.0001	<0.0001
	LAB	1.92 ^Dc^	2.57 ^Cb^	3.04 ^Bbc^	3.36 ^Ac^				
	EN	2.46 ^Db^	2.95 ^Ca^	3.90 ^Ba^	4.33 ^Aa^				
	LAB + EN	1.71 ^Dd^	2.14 ^Cc^	2.81 ^Bc^	3.14 ^Ac^				
LAB (log_10_ cfu/g FM)	Control	7.88 ^Ac^	7.65 ^Bc^	7.39 ^Cc^	7.25 ^Dc^	0.03	<0.0001	<0.0001	<0.0001
	LAB	8.41 ^Aa^	8.26 ^Ba^	8.15 ^Ca^	8.04 ^Da^				
	EN	8.25 ^Ab^	8.11 ^Bb^	7.96 ^Cb^	7.90 ^Db^				
	LAB + EN	8.35 ^Aab^	8.23 ^Ba^	8.15 ^BCa^	8.08 ^Ca^				
Yeast (log_10_ cfu/g FM)	Control	3.15 ^Da^	3.74 ^Cb^	5.69 ^Bb^	7.93 ^Ab^	0.02	<0.0001	<0.0001	<0.0001
	LAB	2.70 ^Dc^	2.91 ^Cc^	3.51 ^Bc^	3.68 ^Ac^				
	EN	3.08 ^Db^	4.08 ^Ca^	6.44 ^Ba^	8.64 ^Aa^				
	LAB + EN	2.71 ^Bc^	2.71 ^Bd^	2.71 ^Bd^	2.75 ^Ac^				
Mold (log_10_ cfu/g FM)	Control	2.15 ^Da^	2.80 ^Cb^	3.25 ^Bb^	4.39 ^Ab^	0.03	<0.0001	<0.0001	<0.0001
	LAB	1.92 ^Bc^	1.97 ^Bc^	1.96 ^Bc^	2.04 ^Ac^				
	EN	2.08 ^Db^	2.94 ^Ca^	3.44 ^Ba^	4.79 ^Aa^				
	LAB + EN	1.92 ^Ca^	1.95 ^Bc^	1.96 ^Bc^	2.09 ^Ac^				

^a–d^ Means in the same column followed by different letters differ significantly (*p* < 0.05). ^A–D^ Means in the same row followed by different letters differ significantly (*p* < 0.05). ^1^ DM, dry matter; LA, lactic acid; AA, acetic acid; AN, ammonia nitrogen; TN, total nitrogen; LAB, lactic acid bacteria; FM, fresh material. ^2^ Treatments, control, no additive; LAB, lactic acid bacteria; EN, 1 g fibrolytic enzyme per 1 kg FM; LAB + EN, combine lactic acid bacteria with 1 g fibrolytic enzyme per 1 kg FM. ^3^ SEM, standard error of mean. ^4^ T, treatment; D, Days of aerobic exposure; T × D, the interaction between treatment and days of aerobic exposure.

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
