# Peer review of "Silage Fermentation, Bacterial Community, and Aerobic Stability of Total Mixed Ration Containing Wet Corn Gluten Feed and Corn Stover Prepared with Different Additives"

_animals, 2020, doi:10.3390/ani10101775_

Round 1

Reviewer 1 Report

How do you establish the TMR ingredient procents?

What represent - concentrate - in TMR content? - Please detail

Line 93 - how do you obtained 21 bags for each treatment? Please detail in squares like you do for the other numbers.

Lines 161-162 - Please verify the sentence ,, Total mixed ration had a low (< 5 lg cfu/g of FM) LAB count and a high yeast count (> 5 lg cfu/g of FM)." by comparing with table 1.

Line 169 - correct the word - Silages - from the subtitle

Did you study the chemical composition, microbial counts and fermentation quality of the corn silage that represent 30.9% of the TMR dry matter? If you did, please include the results in tables. It will be interesting to evaluate the corn silage contribution on TMR chemical composition, microbial counts and fermentation quality.

Author Response

How do you establish the TMR ingredient procents?

Response: Thank you for your suggestion. Ingredient percent of TMR was formulated to meet the animals’ requirements for nutrition based on Cornell-Penn-Miner Dairy model version 3.08.01. The words had been added in new manuscript (Lines 87-88).

What represent - concentrate - in TMR content? - Please detail

Response: Thank you for your suggestion. The content of TMR has been detailed in the new manuscript (Lines 85-86).

Line 93 - how do you obtained 21 bags for each treatment? Please detail in squares like you do for the other numbers.

Response: Thank you for your suggestion. The relative information had been added in the new manuscript (Lines101).

Lines 161-162 - Please verify the sentence ,, Total mixed ration had a low (< 5 lg cfu/g of FM) LAB count and a high yeast count (> 5 lg cfu/g of FM)." by comparing with table 1.

Response: Thank you for your suggestion. The sentence had been reworded in new manuscript (Lines169-172).

Line 169 - correct the word - Silages - from the subtitle

Response: Thank you for your suggestion. This word had been corrected in new manuscript (Lines 179).

Did you study the chemical composition, microbial counts and fermentation quality of the corn silage that represent 30.9% of the TMR dry matter? If you did, please include the results in tables. It will be interesting to evaluate the corn silage contribution on TMR chemical composition, microbial counts and fermentation quality.

Response: Thank you for your suggestion. We believe that fermentation indicators and microbial indicators are not important. According to the studies of Nishino (2015), who believed that “The finding that bacterial community of the TMR silage appeared unrelated to those of ingredient silage”. But we measured the chemical composition and microbial counts. The chemical composition and microbial counts of the corn silage had been added in new manuscript (Table 1, Lines 167-168).

Nishino N , Ogata Y , Han H , et al. Identification of bacteria in total mixed ration silage produced with and without crop silage as an ingredient[J]. Animal science Journal, 2015, 86(1).

Reviewer 2 Report

This is a well thought-out mss, but it needs improvement in English and in data presentation.

line 59 What is VFA? by  'theoretical' do you mean 'approximate'?

90 thorough

94 no decimals needed

125 shaken. what is cabesus? is it cotton?

127 good deposit???

113 mold-- what organisms? what makes up the mold mat?

158 shown. The word 'showed' appears 27 times. Use 'had' or 'were' or something else. Data cannot 'show' anything; they are inanimate.

170 additions

Table 2 Fix the bold DM and lines over and under it.  Have treatments going down, and variables across. No need for SEM or p-value

179 row

220 Item--> variable. Again, switch to days going down the table and variables across

255 what is alpha diversity?

Table 4 needs more explanation as to what the data mean

Figure 1 all the labels are too small-- increase the font size. not clear what the Venn diagram means

267 NGS??

Figs 2 and 3 -- increase font size on titles

Fig 3 not well explained in text.

332 control with a small 'c'-- always-- fix this throught the mss

303 not 'raw' -- row!

390 drop "A study revealed that"

411 drop " It is wellknwn that"

444 that can result

"In this study" appears 9 times-- can remove the phrase without changing the meaning of the sentence

Author Response

This is a well thought-out mss, but it needs improvement in English and in data presentation.

Response: Thank you for your suggestion. English and data presentation were improved in the manuscript.

line 59 What is VFA? by  'theoretical' do you mean 'approximate'?

Response: Thank you for your suggestion. This abbreviation had been defined and spelled out in new manuscript (Lines 632). It’s means approximate. The sentence had been reworded in new manuscript (Lines 84).

90 thorough

Response: Thank you for your suggestion. The mistakes had been corrected in new manuscript (Lines 97).

94 no decimals needed

Response: Thank you for your suggestion. The mistakes had been corrected in new manuscript (Lines 102).

125 shaken. what is cabesus? is it cotton?

Response: Thank you for your suggestion. Cabesus is a cotton fabric. The word had been corrected in new manuscript (Lines 133).

127 good deposit???

Response: Thank you for your suggestion. The words had been corrected in new manuscript (Lines 135).

113 mold-- what organisms? what makes up the mold mat?

Response: Thank you for your suggestion. Mold is a type of fungus, which will be produced when aerobic spoilage occurs during silage storage. the mold mat contains many types of mold such as Fusarium and Alternaria species, Aspergillus flavus and Aspergillus parasiticus. Mold can produce mycotoxins, which endanger animal health, the safety of animal-derived foods, and cause production and economic losses. Mold is a very important indicator of silage fermentation.

158 shown. The word 'showed' appears 27 times. Use 'had' or 'were' or something else. Data cannot 'show' anything; they are inanimate.

Response: Thank you for your suggestion. The word had been corrected in new manuscript.

170 additions

Response: Thank you for your suggestion. The mistakes had been corrected in new manuscript (Lines 180).

Table 2 Fix the bold DM and lines over and under it.  Have treatments going down, and variables across. No need for SEM or p-value

Response: Thank you for your suggestion. The mistakes had been corrected in new manuscript (Table 2).

179 row

Response: Thank you for your suggestion. The mistakes had been corrected in new manuscript (Lines 187).

220 Item--> variable. Again, switch to days going down the table and variables across

Response: Thank you for your suggestion. I tried to modify the comments you mentioned, but I cannot fully understand what you mean. If you have any comments, please feel free to contact us, we will make further changes based on your comments (Table 3).

255 what is alpha diversity?

Response: Thank you for your suggestion. Alpha diversity index is a quantitative measure that reflects how many different species. The value of Alpha diversity index increases both when the number of species increases and when evenness increases.

Table 4 needs more explanation as to what the data mean

Response: Thank you for your suggestion. OTU (operational taxonomic units) is in phylogenetic research or population genetics research, in order to facilitate analysis, the same symbol is artificially set for a certain taxon (line, species, genus, group, etc.). In microbial diversity analysis, all sequences are divided into OTU according to different similarity levels. Generally, if the similarity between sequences is higher than 97% (level), it can be defined as an OTU. Each OTU represents a species. Chao1 and Ace indexes is a measure of species richness. The coverage value refers to the coverage of each sample library. The higher the value, the higher the probability that the sequence in the sample will be detected. This value can reflect whether the sequencing result represents the true situation of the sample. The closer the value is to 100%, the more consistent the sequencing results are with the actual conditions of microorganisms in the sample. Simpson and Shannon indexes are comprehensive indexes reflecting the uniformity and richness of each species in the sample.

Figure 1 all the labels are too small-- increase the font size. not clear what the Venn diagram means

Response: Thank you for your suggestion. The font size had been increased in new manuscript (Figure 1). The Venn diagram shows the number of species unique to the latter among different groups. Each circle represents a group. The larger the area, the more species.

267 NGS??

Response: Thank you for your suggestion. This abbreviation had been defined and spelled out in new manuscript (Lines 338).

Figs 2 and 3 -- increase font size on titles

Response: Thank you for your suggestion. The font size had been increased in new manuscript.

Fig 3 not well explained in text.

Response: Thank you for your suggestion. The sentence had been corrected in the new manuscript (Fig 3).

332 control with a small 'c'-- always-- fix this throught the mss

Response: Thank you for your suggestion. The mistakes had been corrected in new manuscript.

303 not 'raw' -- row!

Response: Thank you for your suggestion. The mistakes had been corrected in new manuscript (Lines 405).

390 drop "A study revealed that"

Response: Thank you for your suggestion. Thank you for your suggestion. The words had been dropped in new manuscript.

411 drop " It is wellknwn that"

Response: Thank you for your suggestion. The words had been dropped in new manuscript.

444 that can result

Response: Thank you for your suggestion. The word had been added in new manuscript.

"In this study" appears 9 times-- can remove the phrase without changing the meaning of the sentence

Response: Thank you for your suggestion. The words had been removed in the new manuscript.

Reviewer 3 Report

Suggestions for Authors

The Authors have investigated an interesting and novel topic and the theme has been properly described. I would like to congratulate Authors for the good-quality of this review article, the literature reported used to write theit paper, and for the clear and appropriate structure.

The manuscript is well written, presented and discussed, and understandable to a specialist readership. In general, the organization and the structure of the article are satisfactory and in agreement with the journal instructions for authors. The subject is adequate with the overall scope of Animals journal.

The work shows a conscientious study in which a very exhaustive discussion of the literature available has been carried out. The introduction provides sufficient background, and the other sections include results clearly presented and analyzed exhaustively. In order to further improve the paper's quality, I recommend to Authors to add these two papers should be added in the Introduction or Discussion section:

Cazzato, E., Laudadio, V., Corleto, A., & Tufarelli, V. (2011). Effects of harvest date, wilting and inoculation on yield and forage quality of ensiling safflower (Carthamus tinctorius L.) biomass. Journal of the Science of Food and Agriculture91(12), 2298-2302.

So, I think that the paper merits the acceptance after very few revisions.

Author Response

In order to further improve the paper's quality, I recommend to Authors to add these two papers should be added in the Introduction or Discussion section: Cazzato, E., Laudadio, V., Corleto, A., & Tufarelli, V. (2011). Effects of harvest date, wilting and inoculation on yield and forage quality of ensiling safflower (Carthamus tinctorius L.) biomass. Journal of the Science of Food and Agriculture, 91(12), 2298-2302. So, I think that the paper merits the acceptance after very few revisions.

Response: Thank you for your suggestion. This reference had been added in the Introduction or Discussion section in new manuscript (Lines 61,442,591).